# Regulation of corneal epithelial differentiation: miR-141-3p promotes the arrest of cell proliferation and enhances the expression of terminal phenotype

María Teresa Ortiz-Melo [1,2], Jorge E. Campos[3], Erika Sánchez-Guzmán[1], María Esther Herrera-Aguirre[1], Federico Castro-Muñozledo[1]*

1 Departamento de Biología Celular, Centro de Investigación y de Estudios Avanzados del Instituto Politécnico Nacional, México City, México, 2 Unidad de Investigación en Biomedicina (UBIMED), Facultad de Estudios Superiores Iztacala, Universidad Nacional Autónoma de México, Tlalnepantla, Estado de México, México, 3 Unidad de Biotecnología y Prototipos (UBIPRO), Facultad de Estudios Superiores Iztacala, Universidad Nacional Autónoma de México, Tlalnepantla, Estado de México, México

* federico.castro@cinvestav.mx, f.castromunozledo@gmail.com

**Data Availability Statement:** The raw FASTQ files and DE results obtained in this work have been deposited in the Gene Expression Omnibus (GEO),

## Abstract

In recent years, different laboratories have provided evidence on the role of miRNAs in regulation of corneal epithelial metabolism, permeability and wound healing, as well as their alteration after surgery and in some ocular pathologies. We searched the available databases reporting miRNA expression in the human eye, looking for miRNAs highly expressed in central cornea, which could be crucial for maintenance of the epithelial phenotype. Using the rabbit RCE1(5T5) cell line as a model of corneal epithelial differentiation, we describe the participation of miR-141-3p as a possible negative regulator of the proliferative/migratory phenotype in corneal epithelial cells. The expression of miR-141-3p followed a time course similar to the differentiation-linked KRT3 cytokeratin, being delayed 24–48 hours relative to PAX6 expression; such result suggested that miR-141-3p only regulates the expression of terminal phenotype. Inhibition of miR-141-3p led to increased cell proliferation and motility, and induced the expression of molecular makers characteristic of an Epithelial Mesenchymal Transition (EMT). Comparison between the transcriptional profile of cells in which miR-141-3p was knocked down, and the transcriptomes from proliferative non-differentiated and differentiated stratified epithelia suggest that miR-141-3p is involved in the expression of terminal differentiation mediating the arrest of cell proliferation and inhibiting the EMT in highly motile early differentiating cells.

## Introduction

Cell differentiation consists of a sequence of coordinated events that result in differential gene expression and culminate in the appearance of specific cell lineages. This process ranges from the establishment of developmental patterns and complex structures [1–3] to the expression of specialized phenotypes [4–7].

National Center for Biotechnology Information (NCBI), and they are accessible through the accession numbers GSE147779 and GSE264014.

**Funding:** This work was supported in part through grant number CF-2022-320450 (to FCML), from Consejo Nacional de Humanidades, Ciencia y Tecnología (CONAHCyT). The funders had no role in study design, data collection and analysis, decision to publish, or preparation of the manuscript.

**Competing interests:** The authors have declared that no competing interests exist.

An example of such complex regulation is the development from cephalic ectoderm, of three contiguous lineages with structural similarities and common regulatory pathways: the epidermal, conjunctival and corneal epithelia [8].

For a long time, transcription factors have stood out as the main protagonists that control cellular programming. For instance, in corneal epithelium, tissue-specific gene regulation has been partially characterized, leading to suggest that transcription factors such as OVOL2, FOXC1, PAX6, and KLF4 are critical regulators of the differentiation program [8–13]. Nevertheless, in recent years, different laboratories provided evidence about the participation of miRNAs as regulators of epithelial cell proliferation and differentiation. In epidermal keratinocytes, miRNAs such as miR-125b and miR-21 support proliferation and suppress differentiation [14, 15]. In contrast, while miR-17 [16] promotes the onset of keratinocyte differentiation, miR-135b [17] and miR-203 [18] regulate the expression of the terminal phenotype.

In comparison with epidermal keratinocytes, miRNA expression in corneal epithelium is less studied. miRNA profiling of basal limbal and corneal epithelial cells from Balb/c mice suggested a significant role of the miR-103/107 family in regulation and maintenance of this epithelium [19]. The importance of these miRNAs was associated with their role in cell communication and adhesion, preserving the integrity of the limbal stem cell niche [19]. Such activity is exerted by targeting specific molecules involved in stem cell renewal, cell cycle regulation, adhesion, and signaling, such as Wnt3a, the ribosomal kinase p90RSK2, the scaffolding protein NEDD9, and the tyrosine phosphatase PTPRM [19, 20]. In addition, comprehensive analysis of the transcriptome and proteome from human limbal epithelium allowed the identification of miRNA-146a as a member of a network that regulates the balance between Notch1 and Notch2, among other signaling processes, to control renewal and differentiation of stem cells [21].

Most miRNAs described for corneal epithelial cells play significant roles during wound healing and migration [20], with exception of miR-145 which suppresses the corneal epithelial progenitor pool, by promoting the expression of terminal differentiation markers such as the K3/K12 cytokeratins and connexin-43, and down-regulating the proliferative/progenitor markers ABCG2 and p63 [22]. More recently, it was described that miR-204-5p participates in maintenance of corneal homeostasis through a co-regulatory mechanism that involves *PAX6* in the regulation of corneal neovascularization and inflammatory response [23].

Based on these previous reports, we looked at the miRNeye database [24] for miRNAs that display high expression levels in central cornea and might be crucial for maintenance of the epithelial cell phenotype [25]. From such preliminary analysis, we found that two miRNAs that belong to the miR-200 family (miR-141-3p and miR-429) [26], and miR-375, a multifunctional regulator involved in pancreatic cell turnover [27] and neuroendocrine differentiation [28], are highly expressed in peripheral and central cornea [24], suggesting their participation in corneal epithelial cell differentiation.

Hence, to explore their possible biological activity in the corneal epithelium, we analyzed its expression in the RCE1(5T5) rabbit corneal epithelial cell line which *in vitro* displays accurately the same sequential developmental stages [13, 29–32] described for primary cell cultures of rabbit corneal epithelial cells [30, 33], and those characteristics found in human corneal tissue [34]. Moreover, this cell culture model has been used both to design a treatment for corneal alkali burns that was successfully used to reduce corneal damage in mice [35], and to assay the damage caused by *Acanthamoeba* on the corneal surface [36]. Using this cell line, we have discerned three different phases in cell culture, which correspond to three stages during the differentiation of the corneal epithelium: i) proliferative, non-differentiated cells that do not express the differentiation-linked KRT3/KRT12 cytokeratin pair and show low or null expression of PAX6 [29, 31, 37]; ii) newly confluent cells, which start the differentiation

program as indicated by the expression of PAX6 [13, 31, 37], and iii) differentiated stratified epithelia, composed by 4–5 cell layers with a structure similar to that found in limbus, with a basal layer which express KRT5/KRT14 cytokeratins, and suprabasal layers expressing the differentiation-linked KRT3/KRT12 cytokeratin pair (about 60–70% of cells) [29, 31, 37].

According to our analysis with TargetScan, miRTarBase, PicTar and miRDB bioinformatic tools, miR-141-3p and miR-429 only possess several target genes in common, despite belonging to the miR-200 family [26]. In contrast with the other two miRNAs, miR-141-3p expression seemed to be related to terminal differentiation. Therefore, we explored its possible role as a regulator of the proliferative/migratory phenotype of corneal epithelial cells. Our results show that miR-141-3p expression follows a time course similar to the expression of the differentiation-linked KRT3 cytokeratin, being delayed 24–48 hours relative to the expression of PAX6, which is suggested as the master transcription factor that drives corneal epithelial differentiation [11, 13, 38]. Also, we show that miR-141-3p inhibition increased cell proliferation and led to the expression of molecular markers related with an Epithelial Mesenchymal Transition (EMT).

## Materials and methods

### Materials

Fetal bovine serum (FBS) was from HyClone Laboratories (Cytiva, Logan, UT). Eagle's medium modified by Dulbecco-Vögt (DMEM) and the Ham-F12 nutrient mixture were from Invitrogen Life Technologies, Inc. (Gaithersburg, MD). TRIzol reagent was from Ambion (Thermo Fisher Scientific, Carlsbad, Ca., USA). All other reagents used were analytical grade.

### Cell culture

The RCE1(5T5) rabbit corneal epithelial cell line was obtained earlier [29]. Cells were plated at $2.7 \times 10^3$ cells/cm$^2$ together with $2.2 \times 10^4$ feeder cells/cm$^2$ mitomycin C-treated 3T3 cells [39], using a (3:1) DMEM/Ham F12- nutrient mixture supplemented with 5%(v/v) FBS, 5 μg/ml insulin, 5.0 μg/ml Transferrin, 0.4 μg/ml hydrocortisone, $2\times10^{-9}$ M triiodothyronine, $1\times10^{-10}$ M cholera toxin, 24.3 mg/l adenine. From the third day after plating, 10 ng/ml EGF was added to the culture medium [29]. Cultures were refed every other day, maintained at 36° C in a 10% CO$_2$ and 90% air-humidified atmosphere.

For cell growth assessment, cells were disaggregated using a (1:1) mixture of 0.15% trypsin and 0.02% EDTA for 20 min at 37°C and quantified using a Neubauer chamber. To establish the number of proliferative cells, we determined the number of colony-forming units in indicator dishes stained with Rhodamine B, which preferentially stains epithelial cells [40].

### Wound healing assay

After feeder removal, 4 day proliferative cultures were transfected either with AntagomiR-141 or the scrambled sequence (see miRNA inhibition). Twenty-four hours after transfection, cultures were trypsinized, cells were harvested and seeded at high densities to obtain confluent epithelia. Then, 48 hours after transfection, the confluent epithelia were wounded using the tip of a Teflon policeman. Wound closure was monitored photographically during 48 hours, evaluating the repopulation of the cleared area during the following 48 hours.

### RNA isolation

After 3T3-feeder cells removal with 0.02% (w/v) EDTA in PBS for 5 min at 37°C, total RNA was isolated with TRIzol® reagent (Thermo Scientific, Carlsbad, CA) at the indicated times, and stored at -70°C as an ethanol precipitate until further use. RNA Integrity and purity were

evaluated using a Bioanalyzer 2100 (Agilent Technologies, Santa Clara, CA); RNA integrity numbers (RIN) of the samples were at least 9.2.

In other experiments, both microRNAs and mRNAs were isolated in fractionated samples (small <200 nt and long > 200 nt RNAs) using the mirVana™ miRNA Isolation Kit (Ambion, Carlsbad, Ca., USA) according to manufacturer's instructions.

## RT-qPCR

RT-qPCR was used to evaluate the expression of selected markers, as previously [37]. For miR-NAs reverse transcription and PCR, we used primers and probes for the TaqMan MicroRNA assay (Ambion, Carlsbad, CA., USA) for miR-141-3p (ID: 000463). As endogenous control, we used snRNA U6B (U6B ID: 001093) (Applied Biosystems, Foster City, Ca., USA). For miRNA reverse transcription, 25 ng from the small RNA fraction (<200 nt) was used along with the MicroRNA Reverse Transcription Kit (Applied Biosystems, Carlsbad, Ca., USA). Once cDNA was synthesized, the PCR reaction was carried out using the TaqMan Universal PCR Master Mix II, No UNG kit (Applied Biosystems, Carlsbad, Ca., USA).

Long RNAs, from the same samples, were subjected to reverse transcription, as described previously [37]. Subsequent PCR was done using the Maxima SYBR Green/ROX qPCR Master Mix (Thermo Scientific, Carlsbad, Ca., USA) and the primers shown in Table 1. PCR reactions

**Table 1. Primer pairs used to determine molecular markers expression through RT-qPCR.**

| Amplified mRNA | Sequence | NCBI access ID or reference | PCR product size (bp) |
|---|---|---|---|
| ΔNp63α | F: CTGGAAAACAATGCCCAGAC | [42] | 196 |
| | R: ATGATGAACAGCCCAACCTC | | |
| CDK3 | F: ACAAGGCCAGGAACAAGGAG | XM_017338904.1 | 118 |
| | R: CCTGACAATGTTGGGGTGCT | | |
| PAX6 | F: CATCTCCCGAATTCTGCAGGTGTC | [13] | 207 |
| | R: CCCTCGGACAGTAATCTGTCTCGG | | |
| KRT3 | F: TCCGTCACAGGCACCAAC | XM_002711005 | 175 |
| | R: TGCGTTTGTTGATTTCGTCT | | |
| KRT12 | F: GTCAGTGTGGAAATGGACGC | XM_004599070.2 | 162 |
| | R: CGTGTTGGTGCTGATCTCCT | | |
| KRT15 | F: ATGTCGAGGCCTGGTTCTTC | XM_008271451.3 | 154 |
| | R: TTCATGCTGAGCTGGGACTG | | |
| VIM | F: GTGTCCTCGTCCTCCTACC | XM_002717420.3 | 266 |
| | R: GTGTTGATGGCGTCGGC | | |
| ZEB1 | F: AAAGGAGCCACAAAAGGACA | XM_017347643.1 | 82 |
| | R: ATTTATGGGGTTGGCACTTG | | |
| ZEB2 | F: AGTGTCAGATTTGTAAGAAAGCG | XM_051848718.1 | 338 |
| | R: GTGCTCCTTCTCGCTCTC | | |
| CDH1 | F: CAATGCTGCCATCGCCTACA | XM_002711639.4 | 109 |
| | R: CCCAGAGGTGACCACACTGAT | | |
| CDH2 | F: AGCTCGTCAGGATCAGGTCT | XM_051851320.1 | 177 |
| | R: GTGCCCTCAAATGAAACCGG | | |
| SNAI1 | F: ACGCTCATCTGGGACTCTCT | XM_051846069.1 | 179 |
| | R: GAGGTGGAGGAGAAGGAGGA | | |
| PRP0 | F: GCAGGTGTTTGACAATGGCAGC | [13] | 231 |
| | R: GCCTTGACCTTTTCAGCAAGTGG | | |

F: forward; R: reverse.

were done in a StepOne™ System (Applied Biosystems, Carlsbad, Ca., USA). Expression was normalized with the $2^{-\Delta\Delta Ct}$ method [41], using the acidic ribosomal phosphoprotein P0 (PR-P0), as internal reference [13, 37]. All PCR quantifications were carried out in triplicates in at least three independent assays.

## miRNA inhibition

For miRNA inhibition experiments, RCE1(5T5) cells were plated as indicated (see above). Four days after plating, feeders were detached from the culture plate by a 0.02% (w/v) EDTA in PBS wash. Thereafter, epithelial cells were transfected with miRVana (Life Technologies, Carlsbad, Ca., USA) antagomiR-141-3p (MH10860) for miR-141-3p inhibition, or with a random scrambled sequence containing the same proportion of purines and pyrimidines as negative control (Cat. No. 4464076). Both sequences were used at 30 nM concentration. Transfection was made with Lipofectamine® RNAiMAX (Invitrogen, Carlsbad, Ca., USA). After 24 hours, fresh media were added and 48 hours after transfection, both small and long RNAs were isolated. The miR-141-3p sequence is highly conserved in human, mouse, rat, rabbit, bovine, goat, horse, chimpanzee and *Pteropus alecto* (black flying fox) (miRBase Access number for rabbit miR-141-3p: MIMAT0048244; miRBase Access number for human miR-141-3p: MIMAT0000432) [43–48].

## Immunofluorescence staining

For immunostaining, cells were grown on $18 \times 18$ mm glass coverslips and fixed at the indicated times. Proliferation was detected with the BrdU assay (Cat. No. 11 296 736 001, Roche Diagnostics GmbH, Mannheim, Germany). Images were captured with a Leica high-speed confocal/multiphoton system (Model TCS SP8-AOBS; Leica Microsystems, Wetzlar, Germany); both xyz and xzy serial optical sections 0.1–0.4 µm thick were taken. To prevent interference from the fluorescent probes, images of the same optical section were taken as separate channels between frames and analyzed with the Leica Application Suite LAS AF v.1.8.0 (Leica Microsystems).

## FACS analysis

Two days after the transfection, cell cultures were disaggregated, and cell suspensions were prepared for flow cytometry [31]. Briefly, $2x10^5$ cells/assay were fixed and permeabilized using BD FACS Perm2 solution (BD Biosciences, SanJose, CA, USA) for 10 min at room temperature. Cells were stained with the indicated antibodies (see Table 2) as follows: Cells were incubated with 5% (v/v) bovine serum albumin in PBS for 10 min at room temperature. After

**Table 2. Antibodies used for immunodetection.**

| Antibody | Antibody specificity | Source | Dilution |
|---|---|---|---|
| Vimentin- FITC (Cat. Sc-6260- FITC) | Monoclonal (V9 clone), recognizes human, rat, pig or birds; fluorescein-conjugated | Santa Cruz Biotechnology Inc. (Santa Cruz CA) | 1:200 |
| KRT3 Cytokeratin | Monoclonal antibody AE5 | Kind gift from Dr. Tung-Tien Sun, New York University | 1:100 |
| Goat anti-mouse Cy5-labeled (Cat. A10524) | Goat-anti-mouse IgG (Cy5 labeled). | Thermo Fisher Scientific (Rockford, IL) | 1:200 |
| Normal mouse IgG1-FITC (Cat. Sc-2855) | Mouse IgG1 Isotype control-FITC labeled. | Santa Cruz Biotechnology Inc. (Santa Cruz CA) | 1:200 |
| Anti-mouse IgG-Alexa Fluor 594 (Cat. A-11032) | Goat anti-mouse Alexa Fluor 594 labeled. Secondary polyclonal antibody. | Thermo Fisher Scientific (Rockford, IL) | 1:200 |

blocking, cells were washed and incubated with the monoclonal antibody AE5 [49] for 20 min on ice and stained with a secondary anti-mouse IgG (Cy5-labeled). After washing with 2% FBS-PBS, cells were incubated with the Vim-FITC antibody for 20 min on ice. Isotype-control matched antibodies (Santa Cruz Biotechnology, Inc.; Dallas, TX) were used as the negative control. All experiments were analyzed with a Cytoflex cytometer (Beckman Coulter, Brea, CA).

### RNA sequencing (RNAseq)

cDNA libraries were generated as previously [37], using a TruSeq RNA Sample Prep Kit (Ilumina, Inc., San Diego, CA). Each library was indexed and pooled. RNA sequencing was carried out at the NGS core facility at the Instituto de Biotecnología, Universidad Nacional Autónoma de México at Cuernavaca, Morelos, México, using an Illumina GAIIx Genome Analyzer-IIx (Illumina, Inc., San Diego, CA) with a configuration for paired-end reads with a 75 bp read length. An average of 7,500,000 reads per sample was obtained. Each sample was considered as replica of the other samples, considering the proportion of constitutive genes. We generated cDNA libraries from proliferative, confluent, differentiated cells [37] and a cDNA library from cells transfected with antagomiR-141-3p.

### Bioinformatics analysis

The quality evaluation of raw reads, the removal of adapters, and the pairing of reads were done using the Geneious workflow (Geneious version 9.1.8) [50]. Paired reads were mapped to the *Oryctolagus cuniculus* genome (OryCun2.0; GCF_000003625.3; 29098 coding genes) using the Geneious for RNAseq mapper as described previously [37].

Differential expression (DE) analysis was done as previously [37], using both the Geneious [50, 51] and VolcaNoseR [52] tools, with a 2-fold change threshold and a p-value < 0.05 to consider a significant difference between compared genes. We also used an Euclidean distance to get the changing and significant top hits for compared genes.

Hierarchical clustering of transcripts per million (TPM) data for both genes and samples was performed with XLSTAT 2017: Data Analysis and Statistical Solution for Microsoft Excel (Addinsoft, Paris, France 2017). Gene ontology enrichment analysis was conducted using Enrichr [53].

### microRNA target gene prediction

For miRNA target prediction, we used the web-based TargetScan 7.1 [54], by using hsa-miR-141-3p, as a searching parameter to select the predicted targets on basis of their cumulative weighted context++ score (CWCS) <0.7.

The targets that were common in both TargetScan and miRTarBase [55, 56], PicTar [57], and miRDB [58, 59] were selected to avoid false positives.

### Statistical analysis

All quantifications were carried out in triplicates in at least three independent assays. Data are presented as the average and their corresponding standard deviations. When appropriate, data were analyzed by using Student's t-test to compare two experimental groups with equal variances, or by using one-way ANOVA with the Holm-Sidak Test for comparison of more than two groups. Statistical significance was accepted if the p-value was lower than 0.05. Statistical testing and plots were performed using SigmaPlot v12.3 software (Systat Software Inc., Chicago, IL).

## Results and discussion

### Time-course of miR-141-3p expression during growth and differentiation of the RCE1(5T5) corneal epithelial cell line

To determine the participation of miRNAs during the growth and differentiation of RCE1 (5T5) cells, we first searched the miRNeye database [24], looking for those miRNAs whose expression was either prevalent at the sclero-corneal limbus or the central corneal epithelium; preferably, with a mutually exclusive distribution. Given that the search at the miRNeye database showed that miR-141-3p was highly expressed in peripheral and central cornea, and considering the results from our preliminary analysis, we chose to study miR-141-3p due to its participation in epithelial [60], neural [61], and BMP2-induced osteoblast differentiation [62]; and because the target genes of miR-141-3p, as predicted by TargetScan 7.1, include markers of both limbal/corneal stem and transient amplifying cells such as p63 [63, 64] or ABCG2 which is considered an specific stem cell marker [65].

We determined miR-141-3p expression during growth and differentiation in cell culture. Starting 3 days after plating, RCE1(5T5) cell cultures were extracted every day for 10 days to quantify miR-141-3p expression by qPCR and compare its time course with the expression of $\Delta Np63\alpha$ as a marker of stem cells and proliferative early differentiating cells [31, 63, 64]; the expression of *PAX6* as a marker of cell programming into differentiation [13]; and K3 cytokeratin (*KRT3*) as an indicator of terminal phenotype [33, 49]. As shown in Fig 1A, miR-141-3p increased significantly in confluent cultures (6th day after plating), to levels 10-fold higher than those found in proliferative cells (Fig 1A). Subsequently, we found levels 40-fold higher in 4–5 layered epithelial sheets (10–12 days after plating) (Fig 1A).

It is noteworthy that the increase in miR-141-3p expression followed the same time-course displayed by the mRNA encoding *KRT3* keratin (Fig 1A), showing a delay of two days about the expression of *PAX6*, which began to increase 4 days after plating to reach a 6-7-fold augment at 8–10 days after plating (Fig 1A). In contrast, $\Delta Np63\alpha$, associated with stem/early precursors/ proliferative cells [42, 63, 64], exhibited high levels in proliferative cells and declined when cultures became confluent and started the differentiation process (Fig 1A). On the other hand, *KRT3* cytokeratin showed a 50-60-fold increase from the $6^{th}$ day in cell culture; at $10^{th}$ day 80% of cells expressed this cytokeratin as previously reported [31, 33]. Since the expression of miR-141 followed the same time-course as the change in the levels of the messenger that encodes K3, being delayed about 48 hours with respect to the increase in Pax6 mRNA, the results suggest that miR-141-3p only participates in the regulation of terminal phenotype expression.

In view of the above and considering previous research which reported that the down-regulation of miR-141-3p promotes EMT and tumor metastasis [66, 67], while its up-regulation inhibits EMT and tumor cell proliferation [67, 68], we also determined the expression pattern of Vimentin (*VIM*) and *ZEB1* along the growth and differentiation of RCE1(5T5) cells. These genes are considered characteristic of the EMT [26, 68] and both were reported as targets of the miR-200 family [69–71]. As shown in Fig 1B, the messengers encoding *VIM* and *ZEB1* showed their highest levels during the exponential growth phase and gradually decreased to reach their lowest levels one day after cells became confluent (Fig 1B). This behavior is consistent with previous results which suggested that proliferating cells undergo the EMT, essential for the highly motile phenotype observed in growing colonies [31]. Interestingly, the decrease in the expression of *VIM* and *ZEB1* started when miR-141-3p levels increased (Fig 1B).

Given these results and considering previous evidence which suggests that members of the miR-200 family, including miR-141, down-regulate EMT [72, 73], it is feasible that miR-141-3p enhances terminal phenotype expression by inhibiting EMT and promoting corneal epithelial cell differentiation.

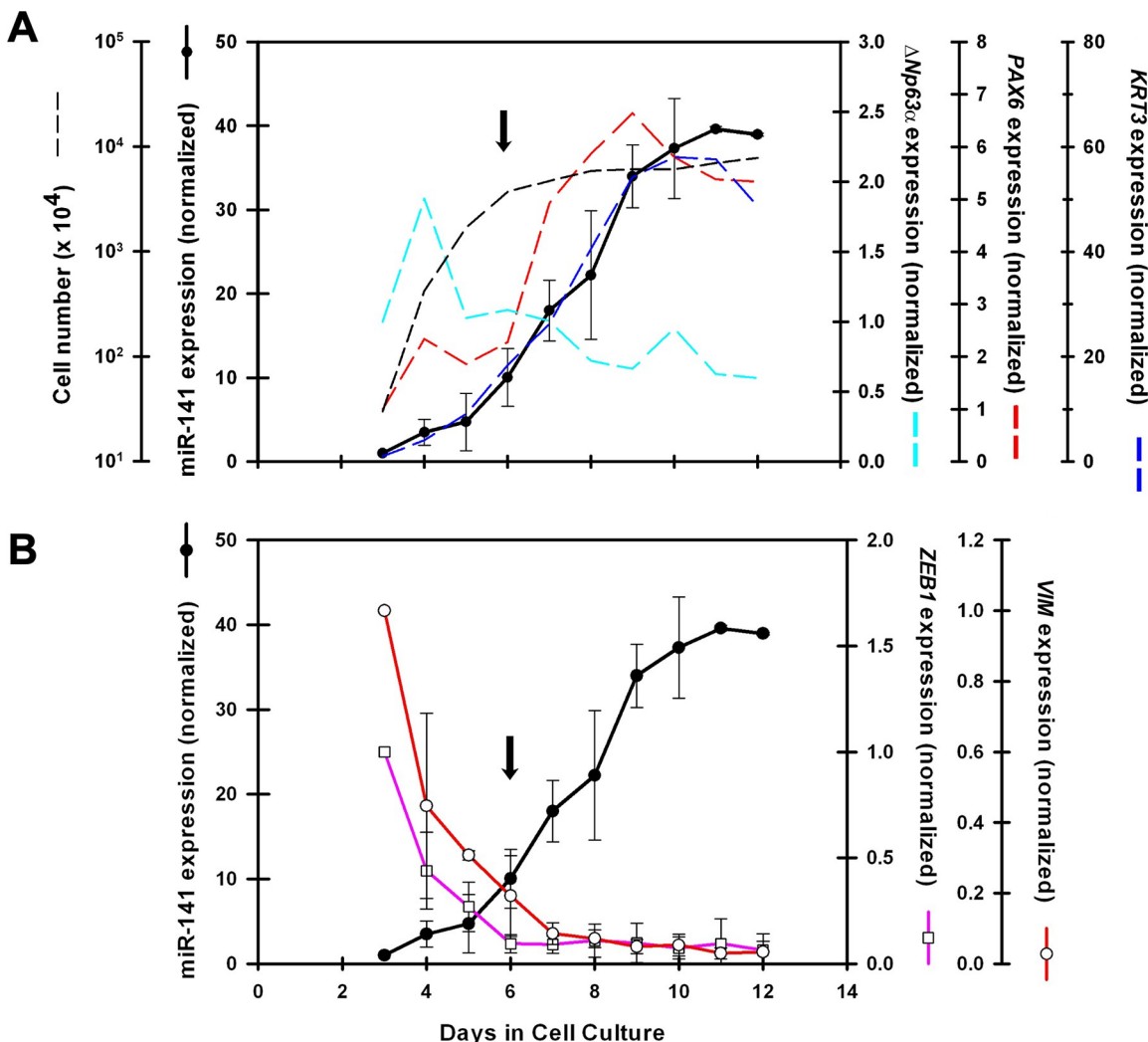

**Fig 1. The spatiotemporal expression of miR-141-3p coincides with the expression of the mRNA encoding keratin *KRT3* during RCE1(5T5) cell differentiation.** After 3 days in culture, total RNA was extracted every day and the expression of **(A)** miR-141-3p was determined by RT-qPCR and compared with the levels of mRNAs encoding the epithelial differentiation markers *ΔNp63α* (cyan dotted line), *KRT3* cytokeratin (blue dotted line), and *PAX6* (red dotted line). **(B)** In the same group of experiments, miR-141-3p expression was compared with the expression of Vimentin (red line), and *ZEB1* (pink line) which has been reported as its specific target. Results correspond to the average of at least 4 experiments carried out with triplicate dishes/day/experiment. The dotted line in (A) shows cell growth (p < 0.05); Note that confluence was reached on the 6th day in cell culture (arrow ↓).

## miRNA-141-3p as regulator of gene expression of RCE1(5T5) cells

In view of the above results, we asked whether inhibition of miR-141 delayed corneal epithelial differentiation by inducing an EMT. To carry out such experiments, we took into account our previous studies showing that 4–6 days after plating, proliferating cell cultures start the differentiation process as determined by the expression of *PAX6* [13, 31, 37]. In addition, on the 6th day in culture, cells showed very low or null levels of stem/transient amplifying cell markers such as *ΔNp63α* [31, 37]. Therefore, 4 days after plating, we transfected proliferating RCE1 (5T5) cells either with 30 nM of AntagomiR-141 or 30 nM of a random sequence containing the same proportion of purines and pyrimidines (scrambled) as the antagonist (Fig 2N); control consisted in non-transfected parallel cultures. Subsequently, 48 hours after transfection

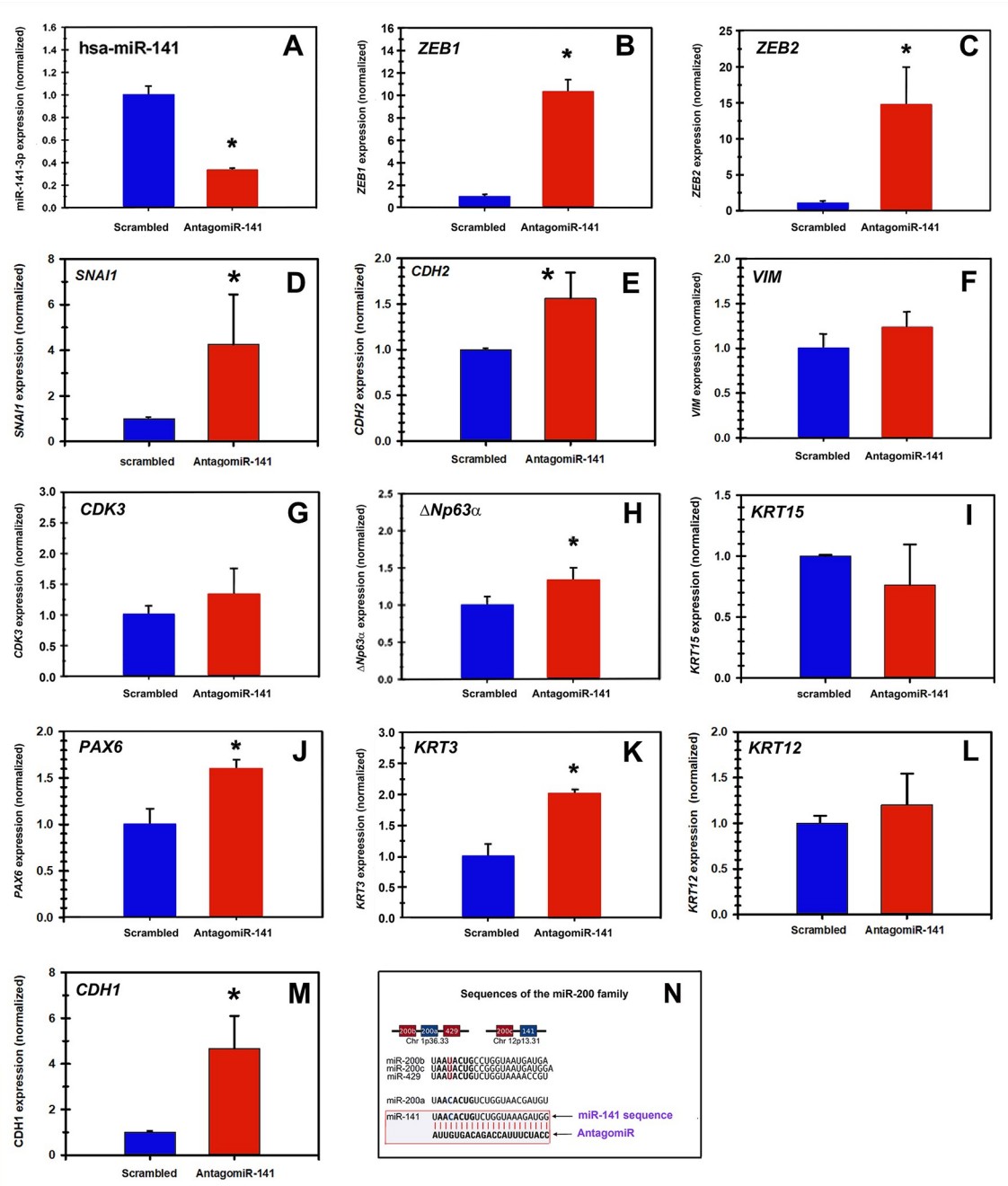

**Fig 2. Inhibition of miR-141-3p results in an increased expression of EMT markers.** RCE1(5T5) cells were transfected either with AntagomiR-141 (red bars) or a scrambled sequence (blue bars). After 48 hours, we measured the expression of EMT and differentiation markers by RT-qPCR. As shown, transfection with AntagomiR-141 led to a 70% reduction in the expression of (A) miR-141-3p and promoted an increase in the expression of (B) ZEB1, (C) ZEB2, (D) SNAI1, (E) CDH2 (N-cadherin), (F) VIM, (G) CDK3 and (H) ΔNp63α, suggesting the induction of an EMT-like phenotype. Interestingly we also detected a significant augment in the expression of (J) PAX6, (K) KRT3 keratin, (L) KRT12 keratin, and (M) CDH1 (E-cadherin). In contrast, we found that (I) KRT15 keratin did not undergo a significant change. Figure shows the results from at least three triplicate experiments (* $p < 0.001$). In (N) we show the sequences of miR-141-3p and its corresponding antagomiR.

(6th day after plating), total RNA was isolated, and the expression of the mRNAs encoding *ZEB1, ZEB2, SNAI1, VIM, N-Cadherin (CDH2), ΔNp63α, KRT15 cytokeratin, E-Cadherin (CDH1), CDK3, PAX6,* and *KRT3/KRT12* cytokeratins were quantified by qPCR. Transfection with the antagomiR led to a 70–80% reduction in the expression of miR-141-3p (Fig 2A), which was accompanied by a 10-fold and 15-fold increase in the expression of *ZEB1* and *ZEB2*, respectively (Fig 2B and 2C), a 4-fold increase in *SNAI1* (Fig 2D), a 50% augment for N-cadherin (Fig 2E), and a 25% rise in the Vimentin encoding messenger (Fig 2F), suggesting that miR-141-3p knocking down favored the expression of an EMT-like phenotype in differentiating cells.

In those cell cultures in which miR-141-3p was inhibited, we also observed a 40% augment in the expression of the cyclin-dependent kinase 3 (*CDK3*) (Fig 2E) which promotes entry into the S phase of the cell cycle [74, 75]; and an increase in the expression of *ΔNp63α* (Fig 2F), proposed as a stem/proliferative cells marker [31, 37, 63, 64, 76]. Taken together, the results suggest that knock-down of miR-141-3p delays the expression of terminal differentiation and promotes a proliferative/migratory phenotype similar to that found in transient amplifying cells.

Nevertheless, knocking-down of miR-141-3p did not inhibit the differentiation process, as indicated by the augment in the expression of *PAX6*, *KRT3* and *KRT12* cytokeratins and *CDH1*, which increased 1.5, 2, 1.2 and 5-fold, respectively (Fig 2J–2M).

## miR-141-3p knock-down leads to the expression of a migratory/proliferative phenotype

To further explore whether miR-141-3p knock-down leads to the expression of an EMT-like phenotype, we assayed the proliferative abilities of cells treated with the AntagomiR-141-3p. Four days after plating, cultures were transfected either with AntagomiR-141 or the scrambled sequence (see Methods), and 48 hours after transfection, cultures were photographed, fixed, and then stained with rhodamine B to determine colony-forming efficiency. As shown in Fig 3A, in AntagomiR-141 transfected cells, we observed an increase in the number of migratory, elongated cells with large lamellipodia, which were located at the migratory/proliferative rim of the growing colonies (Fig 3A, right). In contrast, colonies that received the scrambled sequence were smaller and showed the morphological characteristics described for keratinocyte-growing colonies (Fig 3A left) [77, 78]. Moreover, antagomiR-141-3p led to a 1.4-fold increase in colony-forming efficiency (Fig 3B). These observations are supported by the increase in the expression of the EMT markers described above.

To gain additional evidence showing that miR-141-3p knockdown promotes a migratory phenotype in corneal epithelial cells, we performed wound closure assays. As shown in Fig 3D, transfection with the AntagomiR led to a faster wound closure in comparison with those cells which received the scrambled sequence. Cells transfected with the AntagomiR completely closed the wound within 48 hours; in contrast, control cultures still had uncovered areas after 48 hours (see arrows in Fig 3D)

To confirm whether AntagomiR 141-3p stimulated cell proliferation, we assayed BrdU incorporation in cell cultures transfected with the antagonist. Fig 4A and 4B show that antagomiR-141-3p produced a 30% increase in the number of BrdU-positive cells (Fig 4A and 4B). Together, the above results suggest that miR-141-3p knock-down promotes the expression of a proliferative/migratory phenotype in the corneal epithelial cell cultures.

Previously, it was shown that the main population of proliferative epithelial cells is located at the migratory/proliferative rim both in human epidermal keratinocyte growing colonies and in corneal epithelial cells [31, 78]. Particularly, such cell populations co-express

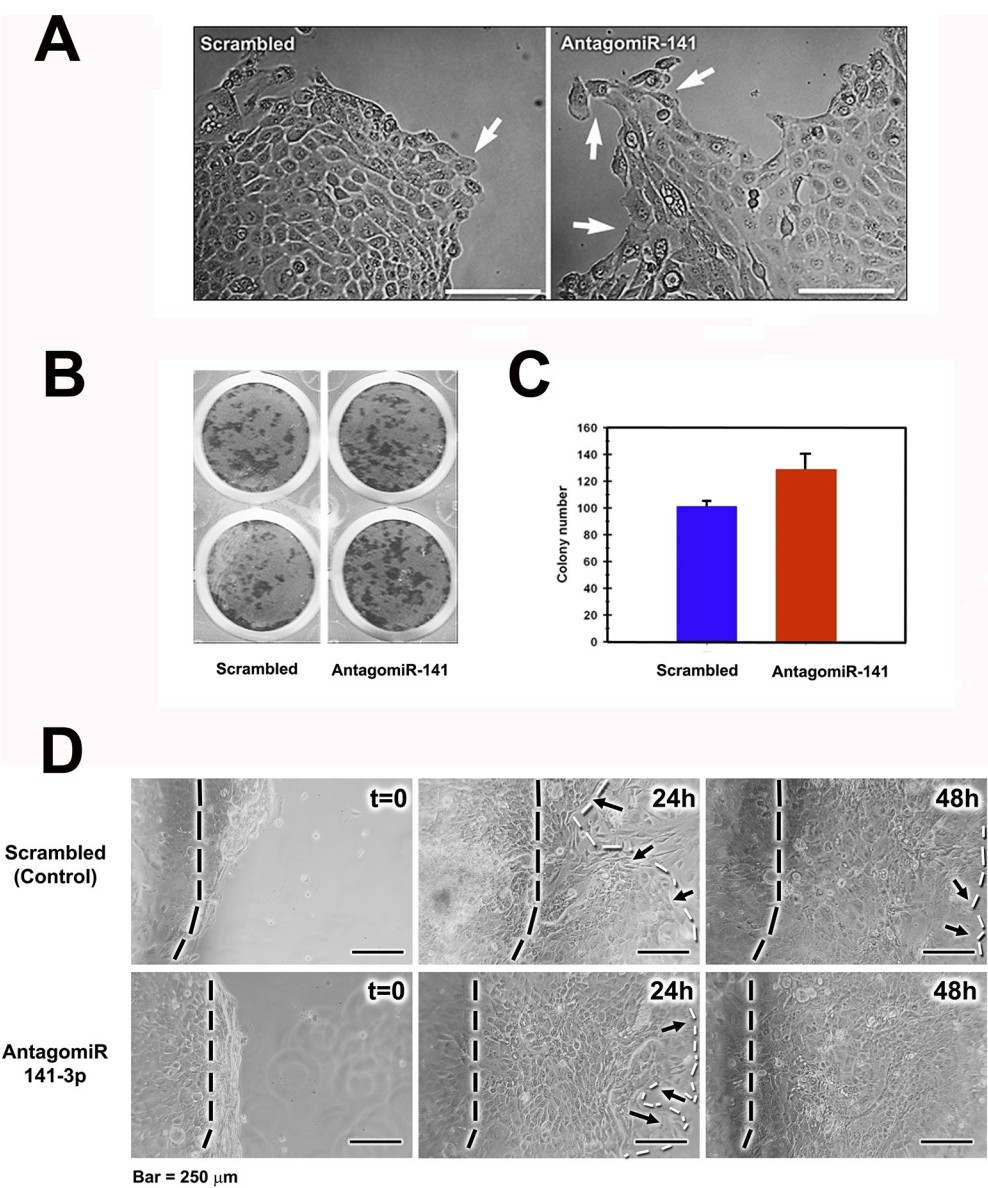

**Fig 3. Inhibition of miR-141-3p promotes a migratory phenotype and leads to an earlier wound closure. (A)** After miR-141-3p inhibition, growing colonies presented a more extended proliferative edge, with a greater number of migratory, elongated cells (arrows), in contrast with the smaller colonies transfected with the scrambled sequence. (Bar = 100μm). **(B)** Such an effect was also observed in the proportion of colony forming cells. **(C)** Transfection with AntagomiR-141 led to an increase, although non-significant, in the number of cell colonies per dish. **(D)** In wound closure assays, cells which received the AntagomiR, closed the wound faster than those transfected with the scrambled sequence, wound closure was complete 48 hours after wounding. Black dotted lines indicate the leading edge of the wound at the beginning of the experiment. White dotted lines indicate the leading edge 24h and 48h after wounding. Arrows indicate the leading edge of migrating epithelium. Bar = 250 μm.

cytokeratin and vimentin filaments, the latter being essential for the expression of the migratory/motile ability of cells [31, 79, 80]. Moreover, our previous report showing that proliferating corneal epithelial cells express a highly motile KRT$^+$/VIM$^+$/PAX6$^{low}$/ΔNp63α$^+$/α6 integrin$^+$ phenotype [31], suggested that these cell populations display an EMT-like phenotype which could be considered a distinctive feature of proliferative and early differentiating cells [31]. Since antagomiR-treated cells underwent an increase in *VIM* and *KRT3* expression

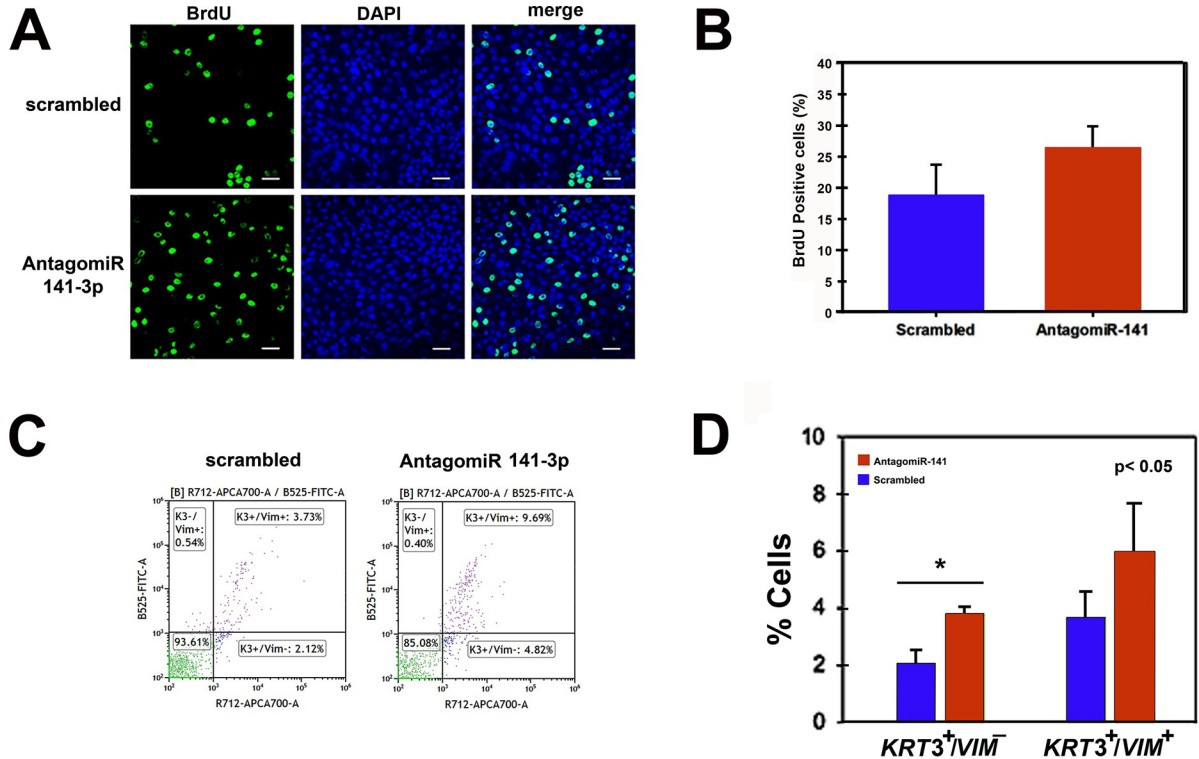

**Fig 4. AntagomiR-141 promotes an increase in both the number of proliferative cells and cells that co-express Vimentin (*VIM*) and the differentiation-linked cytokeratin *KRT3*. (A)** When BrdU incorporation assay was used to determine the number of proliferative cells, it was found that miR-141-3p inhibition promoted a 30% increase in the number of proliferative cells. **(B)** Increase in the percentage of proliferative cells promoted by miR-141-3p inhibition. Bar = 40μm. (n = 3, ± SD). **(C, D)** Cells transfected either with AntagomiR-141 or the scrambled sequence, were double immunostained with antibodies raised against VIM and KRT3. After cytometry, we detected an augment in both the number of cells that co-express VIM and KRT3 and cells that only express KRT3, a result which suggests that inhibition of miR-141-3p promotes a partial EMT in cells that have started the differentiation process.

(Fig 2F and 2K), we wondered if miR-141-3p knocking down promoted EMT in cells located in the early stages of the terminal differentiation process. To explore this scenery, we quantified the number of cells that co-express KRT3 and VIM intermediate filaments by flow cytometry. As observed in Fig 4C and 4D, the knock-down of miR-141-3p promoted a 1.6-fold increase in the number of KRT3+/VIM+ cells and a significant 1.8-fold augment of KRT3+/VIM- differentiated cells (p< 0.05) (Fig 4C and 4D).

Considering that in these experiments, K3 cytokeratin positive cells comprised about 2–4% of cells, and K3/Vimentin positive cells were 4–6% (Fig 4C and 4D), it could be argued that the expression levels of differentiation markers are too low. However, it should be taken into account that the knocking-down of miR-141-3p and quantification of keratin and vimentin expression were carried out during the period comprised between 4 and 6 days of culture, during which the expression of the differentiation process begins.

## Transcriptional profile induced by miR-141-3p inhibition

Previously, we reported the transcriptional profiles of non-differentiated proliferative, confluent, and differentiated corneal epithelial RCE1(5T5) cells [37]. At the same time, we profiled gene expression after knocking down miR-141-3p. As shown in the heatmap (Fig 5A), hierarchical clustering analysis led us to detect 4891 genes differentially expressed between AntagomiR-141-treated cells and these differentiation stages (p-value ≤ 0.01).

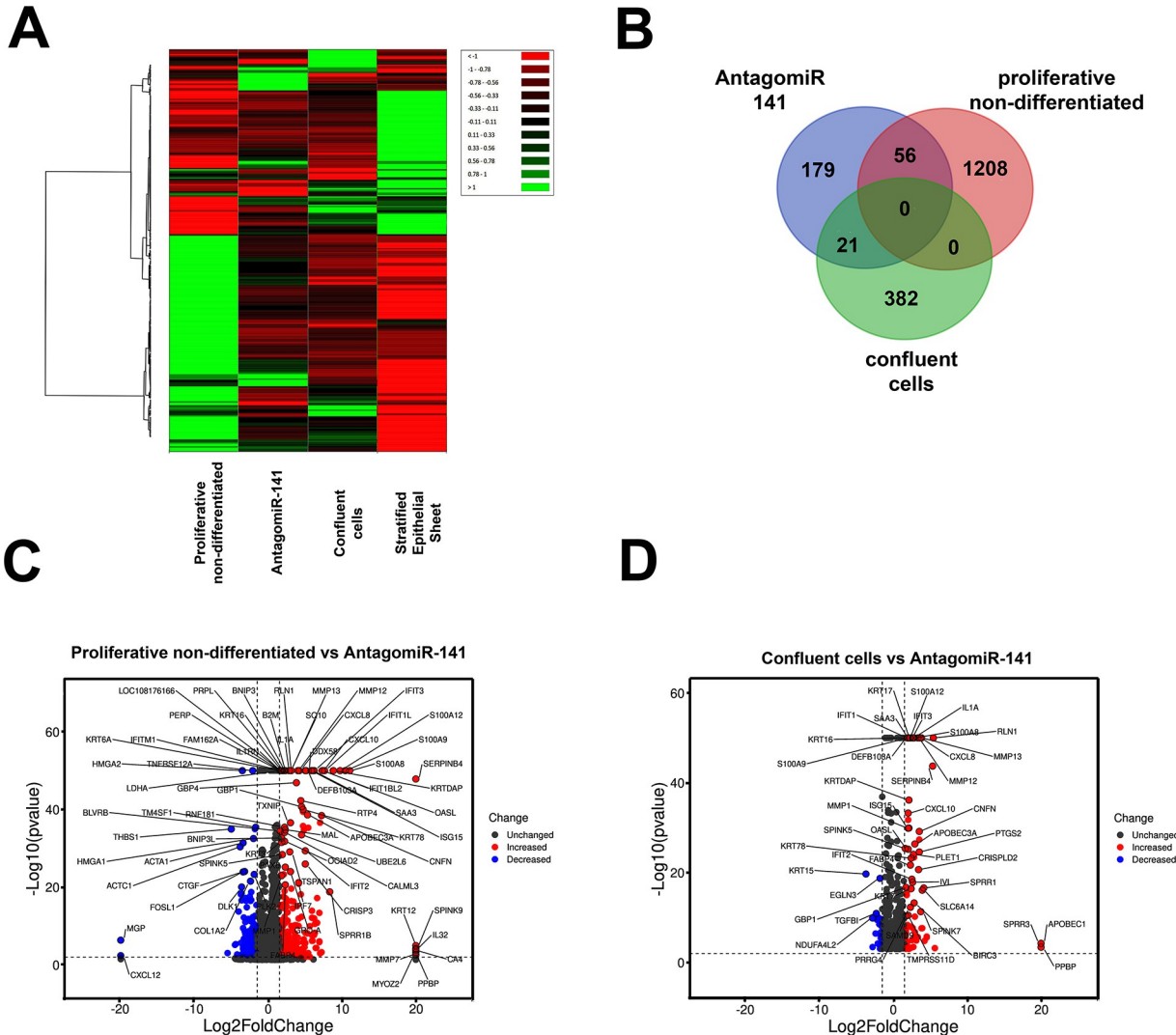

**Fig 5. Transcriptional profile of RCE1(5T5) cells transfected with antagomiR-141. (A)** Heat map that compares the transcriptional profile of cells treated with the antagomiR-141 and cells belonging to each of the three differentiation stages reported previously [37]. From this analysis, we found 4891 genes expressed differentially in AntagomiR-treated cells **(B)** Venn diagram of genes belonging to gene signature of proliferative-non differentiated, confluent, and cells transfected with the antagomiR-141. Volcano plot representations of differentially expressed genes between **(C)** proliferative-non differentiated vs. cells transfected with the antagomiR-141; and **(D)** confluent vs. cells transfected with the antagomiR-141. The x-axes show log2 (Fold-change) and the negative log10(p-value) is plotted in the y-axes. Each point represents a single gene.

In addition, comparison of the gene expression pattern of AntagomiR-141 treated cells with that of proliferative-non differentiated cells revealed that miR-141-3p knock-down led to significant changes in 77 genes (Fig 5C): 63 genes were upregulated, among them those encoding *KRT6*, *KRT16*, and *KRT17* keratins which have been associated with hyperproliferative, altered and transformed epithelia [81–83], as well with wound healing [84]. These changes were accompanied by augments in *MMP1*, *MMP12*, and *MMP13* metalloproteinases, which participate in ECM remodeling and enhance cell migratory/invasive ability [85, 86], associated with EMT [87]. Similarly, compared to the levels found in confluent cells, miR-141-3p knock-down promoted the upregulation of 42 genes (Fig 5D), among these, we found increases in *MMP12* and *MMP13* metalloproteinases.

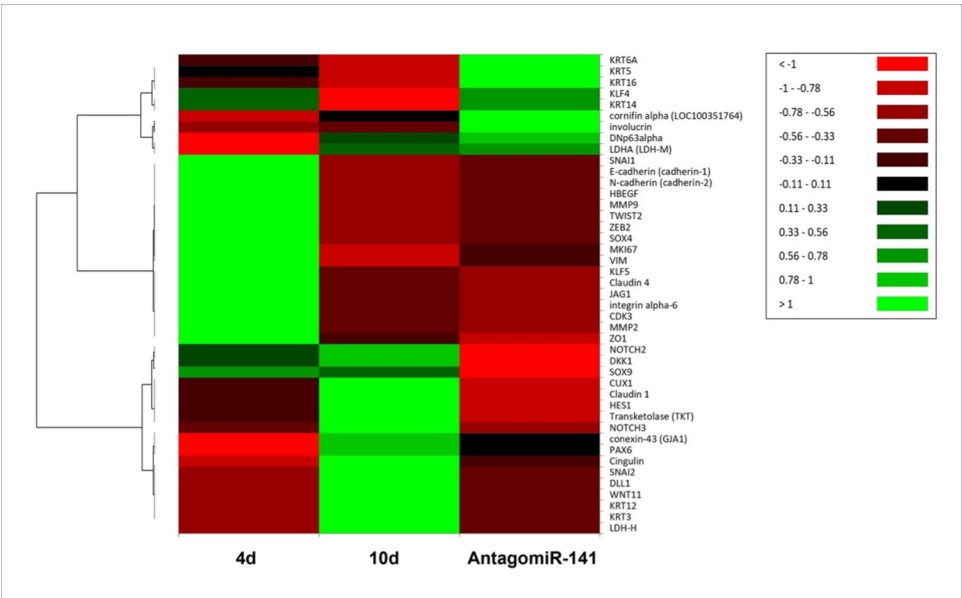

**Fig 6. Expression of molecular markers characteristic of EMT and corneal differentiation in cells transfected with antagomiR-141.** The heatmap shows the comparison between the expression of these genes in transfected cells, with their levels in non-differentiated, proliferative cells (4 days after plating), and differentiated epithelia (10 days after plating). Hierarchical grouping was made using the expression levels in transcripts per million (TPM). Genes that showed a decreased expression are shown in bright red, while those that underwent an increase are depicted in bright green. Note that transcriptomes were specific for each condition.

A deeper analysis, based on hierarchical clustering of selected EMT and epithelial cell markers, showed that overexpressed genes in cells treated with AntagomiR-141 include markers of the EMT-like phenotype such as *SNAI1*, N-cadherin, *HGEGF*, *VIM*, *TWIST2*, and *ZEB2* (Fig 6), and markers associated to proliferating corneal epithelial cells as *MKI67*, *ΔNp63α*, *KLF4*, and the *KRT6*, *KRT16*, *KRT5*, *KRT14* keratins, implying that miR-141-3p inhibition led to an augment in the number of proliferative non-differentiated cells. In contrast, proliferative and differentiated cell cultures showed transcriptional profiles with specific overexpressed clusters that we previously described for each stage of corneal epithelial cell differentiation (Fig 6) [37].

Moreover, we found that the enriched GO categories in AntagomiR-treated cells corresponded to cell proliferation and migration, regulation of cell migration, extracellular matrix disassembly, and pathways such as Hippo signaling associated with EMT (Fig 7A). In contrast, the sets of genes that decreased their expression were those that play a role in the negative regulation of epithelial proliferation and cell migration, as well as genes that regulate ocular development or categories related with negative regulation of the Notch pathway (Fig 7B).

However, despite the increase in cell proliferation and the expression of markers associated with the epithelial-mesenchymal transition, we found that miR-141-3p knock-down was also associated to an increase in the expression of *PAX6* and the differentiation-linked keratin pair *KRT3/KRT12*, as well to a decrease or a non-significant change (as indicated by the qPCR experiments) in *KRT15* keratin levels which has been considered a biomarker of corneal epithelial stem and transient amplifying cells [88, 89]. These results agree with the quantification of some differentiation markers by qPCR (see above), and suggest that miR-141-3p inhibition only delays the expression of terminal phenotype, and simultaneously supports proliferation and the expression of an EMT-like phenotype in cells that started the differentiation process.

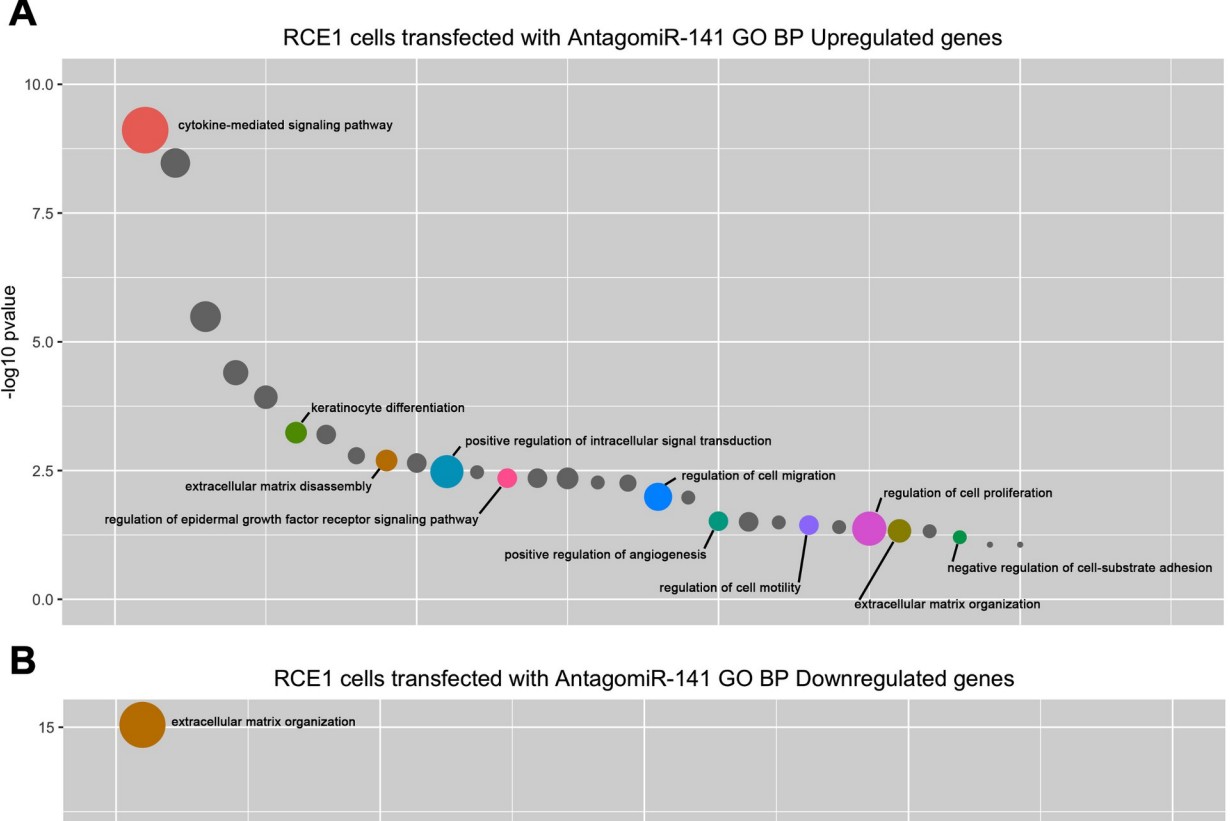

**Fig 7. Gene signature of cells transfected with antagomiR-141. (A)** GO biological processes associated with the signature genes upregulated in cells transfected with antagomiR-141. **(B)** Genes downregulated by antagomiR-141. The y-axis shows the negative log10 of the (p-value) for each term. Circle size corresponds to the number of genes related to the functional category for each signature list. The most representative terms are highlighted in color. Note that overexpressed genes corresponded to categories such as cell proliferation and migration, regulation of cell migration, extracellular matrix disassembly, and pathways such as Hippo signaling associated with EMT.

The decrease in KRT15 expression found when we analyzed the transcriptome from Antago-miR-treated cells, was also detected by qPCR (although non-significant). Such results suggest that miR-141-3p inhibition does not induce a regression to early transient amplifying/precursor cells, and therefore does not block cell differentiation.

As shown above, we detected the upregulation of *ΔNp63α*, *KRT6*, *ZEB1*, *ZEB2*, and *CDK3*; all predicted targets of miR-141-3p (Table 3). Of them, *ZEB1* and *ZEB2* were previously

**Table 3. miR-141-3p selected targets genes.**

| Gene symbol | Gene name |
|---|---|
| ABCG2 | ATP-binding cassette, sub-family G (WHITE), member 2 |
| CDK3 | cyclin-dependent kinase 3 |
| JAG1 | jagged 1 |
| KRT6A | Keratin 6A |
| NOTCH2 | Notch 2 |
| NOTCH3 | Notch 3 |
| NOTCH4 | Notch 4 |
| SP1 | Sp1 transcription factor |
| SNAI2 | snail family transcriptional repressor 2 |
| TP63 | tumor protein p63 |
| TWIST1 | twist family bHLH transcription factor 1 |
| TWIST2 | twist family bHLH transcription factor 2 |
| ZEB1 | zinc finger E-box binding homeobox 1 |
| ZEB2 | zinc finger E-box binding homeobox 2 |

We used the web-based TargetScan 7.1 [54] by using hsa-miR-141-3p, as the searching parameter and selected the predicted targets with a cumulative weighted context++ score (CWCS) <0.7.

reported as targets of the miR-200 family [70], as well as *SNAI2*, *TWIST1*, and *TWIST2* are recognized as EMT activators [90, 91]. Also, the cyclin-dependent kinase 3 (*CDK3*), which regulates cell cycle progression [92]; *KRT6* keratin related to epithelial activation or alteration [93] and *ABCG2* and *ΔNp63α* which are markers of both stem and proliferative/progenitor cells [42, 63, 64, 76], were detected as possible targets of this miRNA. These results imply a crucial role of miR-141-3p, enhancing corneal epithelial differentiation and inhibiting EMT. However, we cannot rule out the participation of the other 4 members of the miR-200 family as cell differentiation regulators, since miR-200a, miR-200b/miR-200c/miR-429, possess seed sequences that only vary from each other by a single nucleotide [69] (Fig 2I), and therefore could share target genes [94, 95], being specific for some others [69]. On the other hand, depending on the epithelial type, these miRNAs can act as EMT repressors as occurs with miR-200a, miR-200b, and miR-200 in PC3 prostate cancer cells [96].

Genomic analysis of miR-200 targets suggests that this family participates in the maintenance of the major characteristics of epithelia, including their integrity. At the same time, these miRNAs inhibit cancer cell motility through their effect on the Rho signaling pathway, and impacti networks that regulate focal adhesion and metalloproteinase activity [97]. However, given the numerous sequences targeted by the miR-200 family, their functions during development and on the expression of different cell types remain unknown. We conclude that miR-141-3p knocking-down mainly affects differentiating corneal epithelial cells. Since miR-141 expression is simultaneous to the expression of *KRT3* cytokeratin, we propose that miR-141 regulates part of the differentiation process by i) promoting the arrest of cell proliferation, and ii) enhancing the expression of terminal phenotype through inhibition of the EMT, as reported for renal cells carcinoma and colorectal cancer [67, 68].

## Conclusion

The miR-200 family comprises five microRNAs that, according to their seed sequences, might be classified into two groups: i) miR-200a and miR-141-3p, with the AACACUG sequence in common, and ii) miR-200b/miR-200c/miR-429, which share the AAUACUG sequence [69].

Although these sequences differ from each other by a single nucleotide, each group possesses specific target genes, enabling gene expression mechanisms that allow the display of different cell abilities. Since miR-141-3p is found in central cornea [24], we explored its biological activity using the corneal epithelial RCE1(5T5) cell line as an experimental model. Our results show that miR-141-3p exhibits the same spatiotemporal expression as the differentiation-linked *KRT3* cytokeratin, implying its role as a positive regulator for the expression of terminal differentiation. Interestingly, miR-200b/miR-429, a member of the second group of the miR-200 family, was reported as an antagonist of corneal wound healing and suppressor of NEDD4 expression, necessary for corneal epithelial cell proliferation and migration [98]. Together, these observations suggest that at least some members of the miR-200 family support the expression of terminal differentiation, perhaps playing complementary functions based on their specificity toward target genes. Our current experiments are aimed to analyze the differential effects of these miRNAs.

## Acknowledgments

We thank Mr. Juan Prado Barajas for his efficient technical help and Mrs. Karina Molina for their secretarial assistance. We also thank Mrs. Luz Rodriguez for her continuous support.

## Author Contributions

**Conceptualization:** Federico Castro-Muñozledo.

**Data curation:** María Teresa Ortiz-Melo.

**Formal analysis:** María Teresa Ortiz-Melo, Jorge E. Campos, Federico Castro-Muñozledo.

**Funding acquisition:** Federico Castro-Muñozledo.

**Investigation:** María Teresa Ortiz-Melo, Jorge E. Campos, María Esther Herrera-Aguirre, Federico Castro-Muñozledo.

**Methodology:** Erika Sánchez-Guzmán, María Esther Herrera-Aguirre.

**Project administration:** Federico Castro-Muñozledo.

**Software:** Jorge E. Campos.

**Supervision:** Jorge E. Campos, Federico Castro-Muñozledo.

**Validation:** María Teresa Ortiz-Melo.

**Writing – original draft:** Federico Castro-Muñozledo.

**Writing – review & editing:** María Teresa Ortiz-Melo, Jorge E. Campos, Federico Castro-Muñozledo.

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
