## [Decision Letter · Decision Letter 0]

16 Sep 2024

PONE-D-24-30995Regulation of corneal epithelial differentiation: miR-141-3p promotes the arrest of cell proliferation and enhances the expression of terminal phenotypePLOS ONE

Dear Dr. Castro-Muñozledo,

Thank you for submitting your manuscript to PLOS ONE. After careful consideration, we feel that it has merit but does not fully meet PLOS ONE’s publication criteria as it currently stands. Therefore, we invite you to submit a revised version of the manuscript that addresses the points raised during the review process.

You will note that the expert reviewers have made substantive recommendations, with which I agree. Both reviewers requested that you justify the use of the rabbit RCE1(5T5) cell line as a model as opposed to human corneal cell lines. Both reviewers requested that you justify the use of KRT3 expression as a marker of differentiation, and reviewer 2 recommended that KRT12 should also be used.  Reviewer 1 points out a contradiction with respect to knock down of miR-141-3p and KRT15, and this should be resolved. Reviewer 2, demands a better characterization of EMT in your model.

We look forward to receiving your revised manuscript.

Kind regards,

Alfred S Lewin, Ph.D.

Section Editor

PLOS ONE

Journal Requirements: When submitting your revision, we need you to address these additional requirements. 1. Please ensure that your manuscript meets PLOS ONE's style requirements, including those for file naming. The PLOS ONE style templates can be found at https://journals.plos.org/plosone/s/file?id=wjVg/PLOSOne_formatting_sample_main_body.pdf and https://journals.plos.org/plosone/s/file?id=ba62/PLOSOne_formatting_sample_title_authors_affiliations.pdf 2. Thank you for stating the following financial disclosure: "The present manuscript was supported in part through grants number 320450 from Consejo Nacional de Humanidades, Ciencia y Tecnología (Conahcyt) from mexican government, granted to Federico Castro-Muñozledo" Please state what role the funders took in the study.  If the funders had no role, please state: ""The funders had no role in study design, data collection and analysis, decision to publish, or preparation of the manuscript."" If this statement is not correct you must amend it as needed. Please include this amended Role of Funder statement in your cover letter; we will change the online submission form on your behalf. 3. Please note that in order to use the direct billing option the corresponding author must be affiliated with the chosen institute. Please either amend your manuscript to change the affiliation or corresponding author, or email us at plosone@plos.org with a request to remove this option.

Reviewers' comments:

Reviewer's Responses to Questions

**Comments to the Author**

1. Is the manuscript technically sound, and do the data support the conclusions?

Reviewer #1: Partly

Reviewer #2: Partly

2. Has the statistical analysis been performed appropriately and rigorously? 

Reviewer #1: N/A

Reviewer #2: Yes

3. Have the authors made all data underlying the findings in their manuscript fully available?

Reviewer #1: Yes

Reviewer #2: Yes

4. Is the manuscript presented in an intelligible fashion and written in standard English?

Reviewer #1: Yes

Reviewer #2: No

5. Review Comments to the Author

Reviewer #1: The authors have explored the role of miR-141-3p in the regulation of corneal epithelial differentiation and demonstrated that it promotes the arrest cell proliferation and enhances the expression of terminal phenotype using rabbit corneal epithelial cell line. The authors need to address the following points of concern for the manuscript to be considered for publication.

1. The authors need to justify the use of rabbit corneal epithelial cell line while there are several human corneal epithelial cell lines. They need to introduce the cell line and their previous work on the same as a model to study the corneal epithelial differentiation. A brief note on the different stages will also enable the readers to understand the significance of the current study.

2. Details on the markers used in the study:

a. K3 – it is a corneal epithelial cell marker expressed at protein level by suprabasal limbal epithelium and central corneal epithelium in humans. In this study, K3 is defined as a terminal differentiation marker. The authors need to justify this as well as specify the percentage of RCE1 (5T5) cells that express K3?

In a primary human limbal explant culture, almost all cells express K3. In contrast, in figure 4A, more than 85% of the cells were negative for K3. To discuss.

b. ΔNp63α – is a corneal epithelial stem cell marker. In has been defined in this manuscript as

i. P63 – epithelial stem cell marker – line no. 262

ii. Marker for proliferative early differentiating cells - line no. 267

iii. Related to early precursor/proliferative cells – line no. 293

iv. Proposed as a stem/proliferative cell marker – line nos. 345,346

v. Proliferating corneal epithelial cell marker – line no. 439

vi. Proliferative/progenitor cell marker – line nos. 484,485

3. The authors specify in line nos. 346-348 that “…knockdown of miR-141-3p promotes the expression of a corneal epithelial precursors-like phenotype..”.

But in line nos. 474-475 they specify “…knock down of miR-141-3p decreased KRT15…. a biomarker for corneal epithelial stem and transient amplifying cells.”

Both are contradictory.

4. Line 289 - … Figure 2G represents PAX6 expression and Figure 2H – KRT3 and not KRT3 and vimentin as specified in the manuscript.

5. Transfection with antagomiR-141 resulted only in 25% increase in vimentin mRNA level which was not significant. Similarly flow cytometry data also indicates only a minor increase in the vimentin expression in cells at protein level. Justify how this is associated with EMT.

Reviewer #2: This study investigates the role of human miR-141-3p in corneal epithelial differentiation using the rabbit RCE1(5T5) cell line. The research provides insights into the miRNA-mediated molecular mechanisms regulating corneal epithelial cell differentiation. However, some issues need to be addressed.

1. The authors use the rabbit RCE1(5T5) cell line as a model, but a justification for choosing this model over human corneal epithelial cells or limbal stem cells would be helpful, as these might more accurately represent human corneal biology. Additionally, it is important to clarify whether the miR-141-3p sequence is fully conserved between rabbits and humans.

2. To comprehensively assess miR-141-3p's impact on corneal epithelial cell differentiation, examining only KRT3 expression is insufficient. KRT12 should also be included, as it is a key marker in conjunction with KRT3 for corneal epithelial differentiation.

3. Regarding the epithelial-mesenchymal transition (EMT) analysis, further evaluation is required. The expression of E-cadherin, a key epithelial marker, should be assessed since its reduction is a hallmark of EMT. Similarly, N-cadherin expression should be measured, as an increase indicates a mesenchymal transition. Additionally, the analysis of Snail or Twist, crucial transcription factors in EMT, would provide more comprehensive insight.

4. To better understand miR-141-3p's impact on the migratory ability of corneal epithelial cells, functional assays such as wound healing or transwell migration should be conducted. Moreover, demonstrating whether overexpression of miR-141-3p can reverse the EMT phenotype induced by its inhibition would provide stronger evidence of its regulatory role.

5. There are areas in the manuscript that could be improved for better clarity and flow:

• The introduction lacks a clearly defined hypothesis or research question. Explicitly stating the research objective would better frame the study.

• In the methods section, related experimental details (e.g., cell culture, RNA extraction, and sequencing) should be grouped into cohesive paragraphs for improved readability.

• The results section could benefit from smoother transitions between experiments. Providing context or linking sentences would help illustrate how the findings build on each other.

• The rationale for selecting specific markers (e.g., KRT3, Vimentin, ZEB1) as indicators of differentiation or EMT should be detailed to clarify their relevance to miR-141-3p's function.

6. Finally, minor grammatical issues, such as sentence structure and punctuation, should be corrected to enhance overall readability.

6. PLOS authors have the option to publish the peer review history of their article (what does this mean?). If published, this will include your full peer review and any attached files.

Reviewer #1: **Yes: **Gowri Priya Chidambaranathan

Reviewer #2: No

---

## [Author Response · Author response to Decision Letter 0]

21 Nov 2024

November 20, 2024.

Alfred S. Lewin, Ph.D.

University of Florida

Section editor.

PLoS ONE

Dear Dr.Lewin:

We are grateful for the valuable comments raised by the reviewers and you, after reviewing our manuscript entitled “Regulation of corneal epithelial differentiation: miR-141-3p promotes the arrest of cell proliferation and enhances the expression of terminal phenotype.” by María Teresa Ortiz-Melo, Jorge E. Campos, Erika Sánchez-Guzmán, María Esther Herrera-Aguirre and Federico Castro-Muñozledo. Based on these comments, we have made modifications, added new results, and carried out text clarifications, as follows:

1. A common concern shared by the reviewers and you, is the use of a rabbit corneal epithelial cell line instead of the use of cells from human origin. To clarify this point, we must remark that our cell line was obtained by serial transfer and avoiding gene manipulation, using an experimental approach similar to that followed by Todaro and Green to generate the 3T3 fibroblast cell line (J Cell Biol. 1963, 17:299-313). This contrasts with those cell lines obtained from human cornea generated by expression of viral sequences (see Araki et al., 1993, Invest Ophthalmol Vis Sci. 34(9):2665-2671, and Kahn et al., 1993, Invest Ophthalmol Vis Sci. 34:3429-3441; Kim et al., 2016, J Pharmacol Toxicol Methods. 78:52-57). The published literature on these cell lines shows that, although they display specific markers of the corneal epithelium, in most cases, their expression pattern is characterized by levels lower than those found in the tissue of origin (see Gipson et al., 2003, Invest Ophthalmol Vis Sci. 44:2496-2506; Rubelowski et al., 2020, Graefes Arch Clin Exp Ophthalmol. 258:565-575; Latta et al., 2022, Graefes Arch Clin Exp Ophthalmol 260:4019-4020). Some of these alterations are associated with the insertion site of those sequences used to generate the immortalized cells. In addition, it was reported that they undergo genomic alterations, becoming heterogeneous cell populations, and in some cases these cell lines are tumorigenic (Yamasaki et al., 2009, Invest Ophthalmol Vis Sci. 50:604-613). 

Other human cell lines have been immortalized by the use of telomerase (Robertson et al., 2005, Invest Ophthalmol Vis Sci. 46:470-478), and more recently, a human cell line was obtained by spontaneous immortalization (Notara and Daniels, 2010, Brain Res Bull. 81:279-286), however, these cell populations have not been extensively characterized. Other cell lines, such as SIRC cells, have an uncertain origin and have been poorly characterized (Leerhoy, 1965, Science, 149:633-634).

In contrast with these cell lines, the RCE1(5T5) cells have been extensively studied, showing a sequential expression of markers similar to that described for primary cultures of rabbit corneal epithelial cells. These cells express the keratin pair K5/K14 as described for primary cultures during early growth, and in basal cells when they have constituted a stratified epithelium (Castro-Muñozledo, 1994, J Cell Sci 107: 2343-2351; Castro-Muñozledo et al., 2017, J Cell Physiol 232(4):818-830): Later, during the exponential growth phase, suprabasal layers express the keratin pair K6/K16, and only when cultures constitute a confluent epithelium suprabasal cells become K3/K12 positive showing similar levels to primary cultures. Such an expression pattern that was previously described, as we said before, is comparable to that found in primary cultures. To illustrate this point, below we have included a figure composed by our results (taken from figure 7 in Castro-Muñozledo, 1994, J Cell Sci 107: 2343-2351) and those obtained by Schermer et al (Fig 2, 1989, Differentiation. 42:103-110). 

Moreover, to our knowledge, the RCE1(5T5) cell line has been fully characterized, on the basis of a day by day expression of differentiation markers, both at protein and mRNA expression, as well as the establishment of tight junctions and development of trans epithelial resistance, and mechanisms that regulate tight junctions’ assembly (Martínez-Rendón et al.,2017, J. Cell. Physiol. 232, 1794-1807; Ortiz-Melo et al., 2013, Biology Open 2:132-143). These cells show an expression pattern of claudins similar to that reported for the corneal epithelium from different species (see Ortiz-Melo et al., 2013, Biology Open 2:132-143). 

This figure depicts, for comparison, the expression patterns of cytokeratins KRT5/KRT14, KRT6/KRT16 and KRT3/KRT12 reported along the growth and differentiation of cultured primary corneal epithelial cell, and RCE1(5T5) cells (LEFT: Schermer et al Fig 2, 1989, Differentiation. 42:103-110, primary cultures rabbit corneal epithelial cells. RIGHT: figure 7 in Castro-Muñozledo, 1994, J Cell Sci 107: 2343-2351, RCE1(5T5) cells.)

Moreover, to our knowledge, the RCE1(5T5) cell line has been fully characterized, on basis to a day by day expression of markers, both at protein and mRNA expression, as well as establishment of tight junctions and trans epithelial resistance, and mechanisms that regulate tight junctions assembly (Martínez-Rendón et al.,2017, J. Cell. Physiol. 232, 1794-1807; Ortiz-Melo et al., 2013, Biology Open 2:132-143). 

On the basis of such analysis, we are confident that the results obtained in RCE1(5T5) cell cultures are representative of what happens in primary cell cultures. Moreover, we have used this cell culture model to design a treatment for corneal alkali burns, which afterwards, was successfully used for treatment of experimental chemical damage in mice (Gulias-Cañizo, et al., 2019, Burns. 45, 398-41), or for in vitro assays for the damage caused by Acanthamoeba on corneal surface (Coronado-Velázquez et al., 2020, Parasitol Int. 74: Art No. 102002).

In addition, we must remark that in the first days after confluence, our model possesses a structure organization that makes it similar to the limbal epithelium, and only after more prolonged periods, the cultured epithelia show K3 keratin expression at the basal cell layer. Since we are interested in the regulation of tissue-specific gene expression, we are confident that our experimental model is ideal for this purpose. In addition, its sequential expression of differentiation markers enables us to analyze such regulatory mechanisms.

Taking into account the above considerations we made modifications to the manuscript as follows:

Reviewer 1.

1. We have modified the introduction to include a paragraph describing our previous work with the cell line, including the requested note which explains the three different stages that we have distinguished in the cell culture system.

2a. We consider that KRT3 keratin is linked to differentiation. Since it is expressed in suprabasal layers of the limbal epithelium, and in both basal and suprabasal cells of central cornea, it could be considered a marker of terminal differentiation due to the loss of proliferative abilities of those cells which express it. However, it may be expressed by cells that have started the differentiation program, which show a limited proliferative ability giving rise to paraclones/meroclones (abortive colonies in cell culture) (see Barrandon and Green, 1987, Proc Natl Acad Sci USA 84(8):2302-2306), and were also described in corneal epithelial cell cultures (Pellegrini et al., 1999, J Cell Biol 145(4):769-782). On the other hand, as requested by the reviewer, we have introduced an explanation about the low percentage of corneal epithelial cells which were detected as KRT3+/Vim+ and KRT3+/Vim- in the experiment that was originally described in Figure 4 (and now moved to Figure 5). The major reason for this result is that flow cytometry experiments as well as all experiments in which miR-141-3p was inhibited with the AntagomiR, were carried out in the lapse comprised between 4 to 6 days, a timing in which cultured cells were starting the expression of the differentiation process. In such sense, we tried to prevent the expression of miR-141-3p during the early stages of cell differentiation since we believe that the inhibition of this miRNA at later times, would not have any effect, since the miRNA began its expression from the fourth day in cell culture, and its rise had a delay of 48 hours in relation to the expression of PAX6. It is probable that the AntagomiR would not have any effect on cells expressing the differentiated phenotype, since at later differentiation stages, targets (at least all the predicted targets) such as ZEB1, ZEB2 and vimentin, would have decreased their expression. (In our experimental model, Vim, ZEB1 and ZEB2 show very low or null expression from the 7th day in cell culture).

2b. As requested, we have corrected the variability in how we defined the role of �Np63� in corneal epithelial cells. 

3. We believe that the expression “corneal epithelial precursor-like phenotype” does not imply that cells are regressing their expression to become precursor cells, since miR-141-3p does not seem to have any target that could lead to “dedifferentiation” of cells. We used such term, in view that some characteristics were similar to those found in proliferative, non-differentiated cells. This fact is supported by our results showing that the transcriptome of AntagomiR treated cells is completely different from the profile of proliferative non-differentiated cells. Moreover, the decrease in KRT15 although non-significant as determined by qPCR, was also decreased when we analyzed the transcriptome of AntagomiR treated cells. Such results suggest that the number of stem cells/progenitors in our experimental model is decreasing and therefore miR-141-3p inhibition does not block differentiation. In this sense, we modified the manuscript to avoid confusion.

4. We have corrected the references to Figure 2, as noted in the manuscript by the reviewer. 

5. Please, refer to answer 2a. It should be noted that differentiation does not occur synchronously. 

Reviewer2.

1. We have modified the introduction to include a paragraph that describes our previous work with the cell line, including the requested note which describes the three different stages that we have distinguished in the cell culture system. Moreover, in this rebuttal letter we justify the use of the rabbit corneal epithelial cell line instead of a human corneal cell line.

Also, as requested by the reviewer, we have included in the miRNA inhibition section a text clarifying the full conservation of the miR-141-3p in human, mouse, rat, rabbit, bovine, goat, horse chimpanzee and in a bat species. 

2. In our previous version of the manuscript we have not included KRT12 keratin since, in rabbit, as well as in human, such keratin shows an 8-10 hour delay in its expression, in relation to the expression of KRT3 keratin. It was previously demonstrated that within the corneal-type keratin pair, during differentiation, KRT3 precedes the appearance of KRT12, both in rabbit and in the chick (Chaloin-Dufau et al., 1990, Cell Diff Dev. 32:97-108) and that in human, the expression pattern of KRT3 is expressed from the 12-13 week of gestation, providing the earliest sign of the corneal epithelial differentiation (Rodriguez et al., 1987, Differentiation 34:60-67). Such results are in contrast with mice tissue, which only expresses KRT12 (Kao, W., 2020, Exp Eye Res. 200:108206). It is considered that in mice, KRT5/KRT12 corresponds to the corneal epithelial-type cytokeratin pair, and for such reason, in mice KRT12 is much more important than KRT3 as a corneal differentiation cell marker. In our case, we expected that KRT12 would be expressed to lower levels than KRT3.

Nevertheless, as requested by the reviewer, we have included the quantification of KRT12 by qPCR, showing that KRT12 has higher expression levels in the cells transfected with the antagomiR-141 (although non-significant), than in control cultures that received the scrambled sequence. (see Fig. 2L).

3. As requested by the reviewer, we determined the expression of N-cadherin (CDH2), Snail and E-cadherin. See figure 2D, 2E, and 2M. The results suggest that effectively miR-141-3p inhibition leads to an EMT. However, since we observed an increase in the expression of PAX6, KRT3, KRT12, and considering that miR-141-3p increases about 48 hours after the expression of PAX6, we think that the increase in EMT and proliferation markers reflects the effect of the miRNA inhibition in the cell population that has started the differentiation process, and only these cells show the EMT, but not those that are more advanced in the differentiation process. This would be an explanation for detecting both an increase in EMT markers and in the differentiation markers (i.e. K3+/Vim+ cells).

4. As suggested by the reviewer, we have included in vitro wound healing experiments, and the results are included in a brand new Figure 3. In addition, we have modified Figure 4, to include some results from the old Figure 3

5. We have modified the manuscript as requested by the reviewer. Moreover, we have grouped related experimental details and made clearer the experimental rational trying to establish an adequate explanation to show the relevance of miR-141-3p in corneal epithelial cell differentiation.

We think that our results could be relevant since the studies about the role of different miRNAs on the corneal epithelial differentiation are scarce. Moreover, we think that these could be of wide interest since they agree with results published by others in other tissue types and developmental processes.

We hope that these modifications made our manuscript suitable to be accepted for publication in PLoS One. 

Thank you in advance, we look forward to hearing from you soon.

Sincerely

Federico Castro-Muñozledo.

Please send all correspondence to:

Federico Castro-Muñozledo Ph.D.

Dept. of Cell Biology

CINVESTAV-IPN.

Apdo. Postal 14-740, Mexico City 07000, Mexico

Phone: 52(55)5061-3985

Fax: 52(55)5061-3393

E-mail: federico.castro@cinvestav.mx

Alternative e-mail: f.castromunozledo@gmail.com

---

## [Editor Report · Decision Letter 1]

25 Nov 2024

Regulation of corneal epithelial differentiation: miR-141-3p promotes the arrest of cell proliferation and enhances the expression of terminal phenotype

PONE-D-24-30995R1

Dear Dr. Castro-Muñozledo,

We’re pleased to inform you that your manuscript has been judged scientifically suitable for publication and will be formally accepted for publication once it meets all outstanding technical requirements. Thank you for addressing the concerns of both expert reviewers thoroughly and for improving the writing style of the manuscript.

Kind regards,

Alfred S Lewin, Ph.D.

Section Editor

PLOS ONE
---

## [Editor Report · Acceptance letter]

27 Nov 2024

PONE-D-24-30995R1 

PLOS ONE

Dear Dr. Castro-Muñozledo, 

I'm pleased to inform you that your manuscript has been deemed suitable for publication in PLOS ONE. Congratulations! Your manuscript is now being handed over to our production team.

Kind regards, 

on behalf of

Dr. Alfred S Lewin 

Section Editor

PLOS ONE